# Task interruptions from the perspective of work functions: The development of an observational tool applied to inpatient hospital care in France The Team'IT tool

**Delphine Teigné**[1,2], **Lucie Cazet**[1], **Guillaume Mabileau**[1], **Noémie Terrien**[1]*

**1** Regional Support Structure (SRA) QualiREL Santé, Hôpital Saint Jacques, Nantes, France, **2** Public Health Department, University Hospital of Nantes, Nantes, France

* nterrien@qualirelsante.com

## Abstract

### Background

In France, hospital units responsible for providing inpatient care have few opportunities to address the issue of task interruptions. In Australia, the Dual Perspectives Method (DMP) has been developed to assess interruptions. The method makes it possible to link teamwork and interruptions, by considering the work functions that constitute the system.

### Objective

To develop a tool that can characterize interruptions from the point of view of work functions that is tailored to French hospital units providing inpatient care. The aim was to adapt the items recorded using the DPM and their response categories, and to study the acceptability of observing interruptions for participating teams.

### Method

The items recorded in the DPM were translated and adapted taking into account the French definition of interruptions. This step identified 19 items that targeted the interrupted professional, and 16 that targeted the interrupting professional. The characteristics of interruptions were recorded in September 2019 among 23 volunteer teams in a region in western France. Two observers simultaneously observed the same professional. Observations lasted seven consecutive hours, and targeted all professional categories within the same team.

### Results

The characteristics of 1,929 interruptions were noted. The observation period was well-received by teams. The following terminology regarding the work functions of the interrupting professional was clarified: "coordination of institutional resources", in relation to "the establishment's support processes", "patient services", and "the patient's social life". We believe that our categorization of response modes is exhaustive.

**Data Availability Statement:** All relevant data are within the paper and its Supporting Information files.

**Funding:** This study is part of the IMPACTT French research project on care system performance (Grant Number: PREPS 18-0047/ AO DGOS). The project was coordinated by University Hospital of Nantes, and funded by the Direction Général de l'Offre des Soins (DGOS; https://solidarites-sante.gouv.fr/ministere/organisation/organisation-desdirections-et-services/article/organisation-de-ladirection-generale-de-l-offre-de-soins-dgos) over the period 2018–2021. The funders had no role in study design, data collection and analysis, decision to publish, or preparation of the manuscript.

**Competing interests:** The authors have declared that no competing interests exist.

**Abbreviations:** DPM, Dual Perspectives Method; HAS, *Haute Autorité de Santé*; ICU, Intensive Care Unit; PMP, Patient Management Program.

## Conclusions

We have developed an observational tool, Team'IT, which is tailored to inpatient hospital care in France. Its implementation is the first step in a system to support teams in managing interruptions, and will enable them to reflect on their working methods, and whether interruptions can be avoided. Our work is part of an approach that seeks to improve and enhance the safety of professional practices, by contributing to the longstanding, complex debate about the flow and effectiveness of patient care.

## Trial registration

ClinicalTrials.gov NCT03786874 (December 26, 2018).

## Introduction

The provision of high-quality healthcare is a worldwide challenge, due to ongoing progress in the field of medicine, an aging global population, and an increase in the number of people suffering from disease [1–3]. Hospitals must think again about cooperation between professionals, and the organization of care pathways, with the aim of maximizing the quality of care provided, while minimizing costs and patient risks [1, 2].

Studies of pathways have adopted various methods and tools in efforts to evaluate professional practices, and analyze and identify patient risks [4]. For example, the concept of the clinical pathway supports the study of key elements of coordination and cooperation that make up the patient's care pathway [1, 4, 5], while process analysis and failure mode analysis are approaches that seek to prevent disruption in the professional activities that are part of the latter pathway [4].

In this context, studies of interruptions adopt a risk management perspective that seeks to investigate disruption in patient care activities and improve the safety of a part of the care pathway [6–9]. It is a topic that has been widely-investigated over the past 15 years. Although interruptions can be necessary in raising alerts and interactions within care processes [6, 10–12], they are also described in the literature as detrimental [6]. Many articles report that interruptions increase the mental and psychological burden for professionals, impact decision-making, lead to information being delayed or forgotten and, ultimately, increase the risk of human error [13–16].

There is no consensus in the literature regarding concepts and definitions of interruptions [17–20]. For example, Speier *et al.* describe interruptions as an identifiable event, the occurrence of which is unpredictable and interferes with the maintenance of attention in a specific task [21]. In France, the following definition, provided by the National Authority for Health (the *Haute Autorité de Santé*, HAS) [22] is used, "[the] unexpected, temporary or definitive cessation of a human activity. The reason can either be specific to the operator, or on the contrary, be external. The interruption disrupts the course of the activity, disturbs the operator's concentration and alters the performance of the intervention. The potential need to carry out secondary activities impedes the completion of the initial activity".

A review of the literature highlights a great deal of heterogeneity in the literature regarding interruptions [6]. Several scientific domains have investigated the topic: epidemiology, healthcare safety, ergonomics, and psychology [6], and different disciplines use different approaches to characterize interruptions and identify solutions [23]. Examples include: continuous video recordings of work processes; qualitative approaches that provide information on

relationships, social dynamics and individual motivation; or a combination of semi-structured interviews and direct observations that examine strategies deployed by professionals to manage interruptions [23].

Most of the literature reports studies that are conducted as close as possible to where care is given, and where there is little margin for error recovery [6]. These studies tend to be compartmentalized; for example, they target specific professions and activities that make up the medical circuit or are performed in the operating theatre [8, 24, 25], and do not look in detail at the determinants of interruptions. Although a mono-disciplinary approach (single profession) can be informative, it does not take into account the interdisciplinary dependencies (between professions) that make up the work system in the healthcare environment [26]. Consequently, interventions that are designed to reduce interruptions are not always effective from the perspective of all stakeholders [27]. Therefore, a major challenge for improving the safety of care is to encourage the team to think about how to better-manage interruptions. This approach would not only strengthen teamwork, but would also enhance job satisfaction, and even performance [28, 29].

In Australia, McCurdie et al. [27] proposed a new approach to observing interruptions in an Intensive Care Unit (ICU). The Dual Perspectives Method (DPM) makes it possible to link teamwork and interruptions, and examines the latter from the point of view of the different work functions that make up the work system. In this context, work functions are what must occur for a work system to fulfill its care objectives. They can be carried out by professionals, or sometimes by intelligent agents [27], and are independent of the worker or his/ her role. The execution of work functions is supported by a set of interconnected tasks, some of which are observable, while others are mental, and are less directly observable. When an interruption occurs, one of the tasks that is carried out by the interrupting professional intersects with one of the tasks that is carried out by the interrupted professional [27]. McCurdie et al. [27] categorize ICU tasks using the four work functions defined by Miller et al. [30]: unit resource coordination, care coordination, patient care planning, and patient care delivery. Furthermore, McCurdie et al. [27] coupled this implementation of the collaborative model of work with questions that were put to interrupted and interrupting professionals. This allowed them to objectify interactions between work functions, and to identify appropriate interventions.

In France, numerous reports highlight the difficulties encountered by healthcare teams in dealing with interruptions. The DPM method appears to be a resource that teams can use to manage interruptions. The conventional hospitalization sector, which is defined as a sector in which a patient's stay in a healthcare facility can be longer than one day, and which requires accommodation for at least one night [31], has received little attention from researchers, despite its importance in the context of safety of care. In 2016, the French government entrusted the country's Regional Support Structures (Structures Régionales d'Appui, SRAs) with the mission of providing support and expertise to hospital healthcare professionals [32] with respect to the quality and safety of care. In this context, one of the country's SRAs (based in the west of France) began work on developing a system to help medical and surgical teams to manage interruptions. The first stage consisted of quantifying interruptions based on observations, and qualifying them from the point of view of work functions.

Thus, the main objective of the research reported in this article was to develop an observational tool to collect examples of interruptions specific to inpatient hospital care in France, which should, in fine, make it possible to characterize them from the point of view of work functions. It was therefore necessary to adapt the information captured, and the response categories proposed by McCurdie et al. [27]. A second aim was to study the acceptability of observations of interruptions by care teams, and examine the feasibility of collecting the opinions of professionals regarding the interruptions that occurred.

Our findings are part of the preliminary phase of the Research Project on the Performance of the Healthcare System (*Projet de Recherche sur la Performance du Système de Soins*), entitled IMPACTT which was implemented by the SRA QualiREL Santé for the period 2018–2021 [33].

## Method

### Definition of the team

A care team is a group of professionals working in a defined unit [28], who interact in the context of the broader functional system. The unit is defined by the specialty of care delivered by the healthcare professionals who make up the team.

The specialties targeted by the IMPACTT project relate to medicine or surgery (excluding intensive care [31]. Care teams are composed of six categories of professions: administrative, logistical and technical, medical, medico-technical and psychosocial, paramedical, and management. Whether the establishment is private or public, these professionals are either employees (physicians, nurses, secretaries, physiotherapists, etc.) or self-employed (mostly physicians).

Professionals were considered to be external to the investigated team either: (*i*) when they only visited the unit in question because they usually worked in another unit in the establishment (porters, radiologists, etc.); or (*ii*) when they worked in the unit in question, but the services they provided (notably accommodation and catering) were the responsibility of a private company [28].

### Inclusion criteria

Inclusion criteria for the teams that participated in the IMPACTT project were: a) being located in the Pays de la Loire region (western France), b) being willing to implement a study on the management of interruptions, and c) being able to receive patients for a stay of more than one day, and requiring accommodation of at least one night, whose care is related to medicine or surgery (excluding intensive care). The establishment's top management (director, director of care) had to commit to encouraging the team to implement the support system developed by QualiREL Santé. Team inclusion was completed in November 2018.

### Development of data collection tools to characterize interruptions

**Characterizing interruptions based on the DPM.**   In the DPM, two data collection guides are provided: A (thread concerning the observed professional) and B (thread concerning the interrupting professional) [27]. Details of these guides were obtained from the corresponding author of [27], and they consist of 13 and 12 pieces of information, respectively. Some items can be observed directly, while others are obtained by asking the observed (interruptee) and interrupting (interrupter) professional. These items provide information on the location of the interruptee and interrupter, how the request was made, and how the interruptee reacted. They also characterize tasks (description, routine, work functions), the interruptee and interrupter (professional role and seniority) and the workload (time remaining to complete the task), along with the duration of the request. Finally, other items target information or planning measures that may have prevented the request. Information related to both A and B is captured each time a new interruption occurs. Table 1 shows the number of items per theme, and per data collection guide, in the original version and in our experimental version.

**Table 1. Adaptation of the items in the data collection guide reported in McCurdie et al. [27].**

| Item theme | | Items in the original version | | Items in the experimental version | | | | |
|---|---|---|---|---|---|---|---|---|
| | | Number | | Number | | Response modes | Collection methods | Feasibility of data collection |
| | | Guide A | Guide B | Guide A | Guide B | | | |
| Location of the interruptee and interrupter | | 1 | 1 | 1 | 1 | free text | Observation Questioning | high low |
| Request mode | | 1 | 0 | 1 | 0 | in person/ telephone/ medical alarm/ self-interruption/ other | Observation | high |
| Reaction of the interruptee | | 1 | 0 | 1 | 0 | ignored the question/ refused to answer/ answered later/answered while continuing the task/paid attention to the answer/ other | Observation | high |
| Time of the request | Duration | 1 | 1 | 1 | 0 | free text | Observation | high |
| | Times | 0 | 0 | 2 | 0 | free text | Observation | high |
| Information or measures that could have prevented the request | | 1 | 4 | 1 | 5 | free text | Questioning | low |
| Characterization of the task carried out by the interruptee and interrupter | Description of the task performed | 1 | 1 | 1 | 1 | free text | Observation Questioning | high low |
| | Description of the task requested | 1 | 0 | 1 | 1 | free text | Observation | high |
| | Planning of the task performed | 0 | 0 | 1 | 1 | planned/ unplanned | Questioning | low |
| | Whether the performed task is routine or not | 1 | 1 | 1 | 1 | routine/ non-routine | Observation | medium* |
| | Classification of the task performed according to the PMP | 0 | 0 | 1 | 1 | admission/ prescription-development of the care plan/care plan delivery/ preparation for discharge/ discharge | Observation | medium* |
| | Classification of the task performed according to work functions | 1 | 1 | 1 | 1 | unit resource coordination, care coordination, patient care planning, patient care delivery | Observation | medium* |
| | Classification of the task requested according to work functions | 1 | 0 | 1 | 0 | | Observation | high |
| Characterization of interruptee and interrupter | Professional role | 2 | 1 | 2 | 1 | administrative/ logistical and technical/ medical/ medicotechnical and psychosocial/ paramedical/ management/ patient (or their entourage)/ other | Observation | high |
| | Seniority | 0 | 1 | 1 | 1 | free text | Questioning | medium* |
| Workload of interruptee and interrupter | Time remaining to complete the task | 1 | 1 | 1 | 1 | plenty of time/ not much time left | Questioning | medium* |
| | Intensity of the activity | 0 | 0 | 1 | 1 | low/ medium/ high | Questioning | medium* |

*'Medium' indicates high feasibility of collecting items in dataset A and low feasibility of collecting items in dataset B.

### Translation and cross-cultural adaptation of the DPM data collection guide

First, the items making up the two DPM data collection guides were translated into French by three native French speakers fluent in English. One translator was familiar with the concepts being measured, while the two others were unfamiliar with the care setting. Any discrepancies

in these translations were discussed by the project team, and French-language pre-experimental versions of the two data collection guides were produced.

The research team extended this pre-experimental version based on its knowledge of French teams in the conventional hospital sector, and the literature. Six items were added to part A, five items were added to part B, and the modalities for answering certain items were specified. Please see the S1 File, for full details of this pre-experimental version.

The resulting version was adopted as the French-language experimental version of the data collection guide. Table 1 gives the number of items in this experimental version by theme. It also presents the method used to collect information (observation or questioning), and expected response modes.

## Implementation of the data collection guide

In the original version, the DPM is implemented simultaneously by two observers. Each observer is provided with the same booklet that he/ she uses to collect data about interruptions. This booklet contains several copies of the two data collection guides. When an interruption occurs, each observer manually records the initial details (part A). Then, one of the two observers records the resumption of the task by the interruptee (part A), while the second observer questions the interrupter to contextualize the interruption (part B). Once these observations are completed, a debriefing session is organized with the interruptee, to collect information about his or her perception of the interruption, and record their views on how the work system could be improved (part A). The two observers' handwritten notes are supplemented, if necessary, by audio recordings. In the DPM, the two observers follow the same professional for three hours. Several professionals are observed on different days. Only interruptions judged as non-trivial and non-routine by observers are taken into account.

In the IMPACTT study, pairs of observers were drawn from the project's coordination unit. Their roles were similar to those in the DPM. They were provided with the same observation booklet, in which up to 50 interruptions could be recorded (one interruption per two pages; part A on the left-hand page and part B on the right-hand page). The HAS definition of an interruption was used [22]. Observers were asked to record all interruptions, regardless of whether they were trivial or routine. For each care team, observations were carried out during a single day for approximately seven hours. Different professionals in different categories were observed for about 30 minutes, and the same professional could be observed several times during the day.

At the end of the observation period, a short debriefing session was organized with the interruptee to record his or her perceptions of the interruption, and their opinion on potential ways to improve the work system (part A). Professionals external to the team, along with patients and their entourage, were included in the data collection exercise from the perspective of the interrupter. With respect to professionals, both interruptees and interrupters were asked to give oral consent on the day observations were carried out. Patients and their entourage were not questioned. Observers did not enter patients' rooms. If necessary, professionals were questioned after leaving the patient's room to allow relevant information to be recorded. In general, it was agreed that interruptees and interrupters should be questioned at an appropriate time in order to avoid being the source of another interruption. All of the observers' notes were handwritten.

Observations were conducted from May to September 2019.

## Retranscription of the data collected

All the information written down by observers was transcribed the following day and recorded in a database developed for this purpose. The pairs of observers were asked to find a consensus in the case of disagreement.

Examples were associated with each of response modes as observations were collected. Data processing was carried out using Excel® version 2007.

## Ethical considerations and consent

The IMPACTT non-interventional research protocol was authorized by the Health Research Information Processing Advisory Committee and received approval from an ethics committee (the Gneds) on January 17, 2019. Information about the project was posted on the premises of the observed teams. Participation (interruptees and interrupters) was conditional on oral consent being given. In accordance with articles L1121-1 and R1121-2 of the French Public Health Code, Institutional Review Board approval was not necessary.

## Results

### Characteristics of the included teams

A total of 23 teams were included. Table 2 presents the measured variables and descriptive statistics. Both private ($n = 7$) and public ($n = 16$) legal statuses were represented, and eight teams worked in establishments with capacity of over 300 beds.

### Implementation of the data collection guides

The experimental version of the two data collection guides was used with all 23 teams. Overall, 286 team members were observed as potentially-interrupted professionals. Each observation lasted an average of 40.9 minutes/ professional. All professional categories were observed and examined from the perspective of both the interruptee and interrupter. Table 2 presents variables and descriptive statistics related to the implementation of the data collection guides.

### Completeness of response modes characterizing interruptions

The observers recorded the characteristics of 1,929 interruptions during interactions between professionals.

Table 2. Variables and descriptive statistics of participating teams and observations.

| Variables describing teams included in 2017 | | n (%) or mean (SD) |
|---|---|---|
| Distribution of teams by legal status | private | $n = 7$ |
| | public | $n = 16$ |
| Number of beds (including day patient care) (excluding care for the elderly or disabled, and home care) | per site | 344 (SD = 309) 8 (> 300 beds) vs. 15 (< 300 beds) |
| | per team | 30.8 (SD = 11.0) |
| Number of salaried and self-employed staff | per establishment | 1,204.2 (SD = 1440.0) |
| | per team (daytime including cross-functional) | 48.5 (SD = 21.2) |
| Variables describing observations | | n (%) or mean (SD) |
| Number of professionals observed per team Number of occupational categories observed per team | | 12.4 (SD = 2.4) 3.9 (SD = 0.8) |
| Number of interruptions observed Number of interruptions observed per team | | 1,929 84.3 (SD = 29.9) |
| Number of interrupting occupational categories observed per team | | 6.6 (SD = 1.1) |
| Duration of observations | per team | 486.6 min (SD = 80.4) |
| | per professional observed | 40.9 min (SD = 10.8) |

Response modes to requests were characterized and categorized. This resulted in the addition of two new request types: those due to a failure or lack of equipment, and those due to a failure or a lack of information. No changes were needed to response modes describing the response of the interruptee, or the stage in the PMP. Similarly, no changes were made to the work functions of the interruptee; on the other hand, four external levels were added to those related to the interrupter: coordination of institutional resources; related to the patient's social life; related to the establishment's support processes; related to patient services. Finally, response modes describing the interruptee and/or interrupter were supplemented with a category that captured simultaneous requests by several professionals.

Table 3 presents, and illustrates with examples, response modes for the two categories of professionals (interruptees and interrupters), interruption types, work functions, and the PMP stages.

All of the information that could be collected by direct observation was recorded in each guide. Information or planning measures that could have prevented the request from being made was provided by the observed professional for 22 interruptions (22/1,929 interruptions; 1.1%). Information in part B, obtained by questioning, concerned 31 instances where professionals were interrupted (31/1,196 interruptions; 2.6%). Table 1 shows the feasibility of data collection (observations and questioning) using the IMPACTT method.

## Discussion

### Main findings

The objective of the present study was to develop a tool that could be used to collect observational data regarding the characteristics of interruptions that affected the work of professionals, in particular, with respect to work functions. The study was made possible by adapting the DPM developed by McCurdie *et al.*, which was first used in Australia with an ICU team [27]. Response modes related to the collection of information about interruptions took into account the French definition of the latter term. Terminology related to work functions was specified in relation to the work system currently used for inpatient hospital care in France. The research team also took the opportunity to add further items to enrich the characterization of interruptions.

The observation period was well-received by all participating professionals (interruptees and interrupters). All of the information that could be directly observed was recorded as the observation progressed. The *a priori* response modes were verified and enriched. Given the number of interruptions that were recorded, from a large number (23) of teams, we have reason to believe that the list of response categories was exhaustive. The set of associated examples is of considerable interest for future observations, and has no precedent in the literature.

A first illustration of the completeness of response modes concerns the typology of interruptions: those due to a failure or a lack of information, and those due to a failure or lack of equipment. These types of interruption have been reported in the literature [34–36], and it is easy to see that a failure to keep trolleys well-stocked (with, for example, consumables) or a lack of communication when the care plan is updated, can be causes of interruptions. The second illustration concerns the work functions of the interrupter ("coordination of institutional resources", in relation to "the establishment's support processes", "patient services", and "the patient's social life"). The team is an open system that is in interaction with, on the one hand, other teams in the establishment or external structures and, on the other hand, the patient's entourage. These new work functions make it possible to categorize, among other things, requests to replace staff, computer breakdowns, or even the activation of a television subscription.

**Table 3. Categorization of participating professionals, task interruptions, work functions and stages in the PMP, with examples.**

| Categorization of interruptees and/ or interrupters | Related examples |
|---|---|
| **A.** Administrative (*, **) | All administrative staff, secretary, Quality manager, including students. |
| **B.** Logistical/technical (*, **) | Electrician, plumber, cook, laundry, maintenance of premises, porter, computer specialist, including students and trainees. |
| **C.** Medical (*, **) | Doctor, pharmacist, midwife, including interns and students. |
| **D.** Medicotechnical/ psychosocial (*, **) | Laboratory technician, social worker, psychologist, social worker, including students and trainees. |
| **E.** Paramedical (*, **) | Nurse, healthcare assistant, physiotherapist, dietician, psychomotor therapist, occupational therapist, speech therapist, including students and trainees. |
| **F.** Management (*, **) | Department head, healthcare manager, administrative manager, other, including students and trainees. |
| **G.** Supplier (**) | Ambulance driver, medical visitor, delivery staff. |
| **H.** Other (**) | Pastoral support, chaplain. |
| **I.** Patient/ their entourage (**) | Patient, patient's family, volunteers, other. |
| **J.** Several categories (**) | See examples from each category. |
| **Categorization of task interruptions** | **Related examples** |
| In person*** | Information about the temporary absence of a member of staff. |
| By telephone*** | Call to find out the availability of beds in the department. |
| Due to a medical alarm*** | Medical alarm. |
| Self-interruption*** | Stopping a colleague to pass on information. |
| Failure or lack of equipment*** | Lack of a dressing in the room; a missing cable to perform an ultrasound in the room. |
| Failure or lack of information*** | Incorrect communication of the room number requiring the planning to be looked at again; request to a doctor to confirm that the patient does, in fact, require additional safeguards. |
| **Categorization of the interrupted and/ or interrupting task** | **Examples of tasks performed by the interruptee or interrupter** |
| Unit resource coordination (a,b) | Physical resources: management and replenishment of stocks (dressings, medicines, drug controls, ECG, scales), orders to ensure the availability of all healthcare products, general restocking of trolleys, cleaning of equipment, disinfecting. Human resources: scheduling. Logistical resources: management of the maintenance of the premises, management of beds, management of the preparation and ordering of meals, linen and repairs. Financial resources: billing for procedures. |
| Care coordination (a,b) | Management and distribution of technical care and support activities, Requests related to the progress of care (need for help or offer of help). Managing unplanned absences. Operational meeting, time to summarize and communicate updates to the care plan (medical and paramedical information, social summary). |
| Patient care planning (a,b) | All organizational activities related to the patient's personal care plan: making appointments with internal and external services (scans, consultations with a psychologist, physiotherapist, intervention by a social worker, organization of the patient's discharge). Preparation of requests for biological tests and scans, prescription, reading and communication of results, management of emergency beds. |

(*Continued*)

**Table 3.** (Continued)

| Categorization of interruptees and/ or interrupters | Related examples |
|---|---|
| Patient care delivery (a,b) | Preparation of medications and dressings, preparation of medication trolleys prior to room rounds, preparation of meal trolleys.<br>Technical interventions (infusions, etc.) and support (help with washing, dressing), and information about the patient's status and care.<br>Transcription and validation of interventions |
| In relation to the patient's social life (b) | Support from the patient's family and friends.<br>Activation of the establishment's services (television, laundry, shopping). |
| Coordination of institutional resources (b) | Physical resources: regulation and coordination of emergencies for the establishment.<br>Human resources: replacement at the institutional level, support for student trainers.<br>Logistical resources: development and monitoring of institutional protocols. |
| In relation to patient services (b) | Television subscription activation. |
| In relation to the establishment's support processes (b) | Repair and checking of equipment (computers, alarms) by ancillary services or by professionals working for the establishment, but external to the team.<br>Contacts with the Admissions Office.<br>Distribution of laundry, mail (messengers). |
| **Categorization of the task performed and/or requested according to the PMP** | **Examples of tasks** |
| Admission | Checking admission to the department, preparation of admitted patient files, consultation of admitted patient files, medical examination of an admitted patient. |
| Prescription—development of the care program | Time to summarize and communicate information (medical, nursing, or paramedical, social summary).<br>Care planning, preparation of requests for biological tests, writing up medical observations, prescribing medications.<br>Ongoing medical treatment (or not), technical interventions, and support. |
| Preparation and delivery of care | Preparation for the distribution of medication, preparation of equipment needed for patient care.<br>Technical interventions and support.<br>Transcription of interventions. |
| Preparation for discharge | Writing letters in preparation for discharge.<br>Communication with an external facility for patient discharge. |
| Discharge | Recording reports of the patient's stay, validation and sending of liaison letters. |

* Observed (interrupted) professional categories

** Professional categories that can be interrupting

*** Interruptions that are part of professional interactions

a Categories of observed (interrupted) tasks

b Categories of tasks that can be interrupting

McCurdie *et al.* [27] also categorized the tasks of interrupters in work functions external to those of the team: 'unit resource coordination' and 'care coordination'. Finally, the latter study subdivided the patient care delivery work function into interruptions related to "administration, diagnosis, monitoring, and prescription". The IMPACTT research team chose not to subdivide this work function. However, these elements could be investigated *via* an additional

item that provided information on the PMP. This item provided us with exhaustive information about the patient's care pathway, beginning with their admission, through preparations for discharge, and then discharge.

Collecting information about the context of an interruption proved to be tricky for both interruptees and interrupters. First, there were a lot of interruptions; second, interruptees rarely stopped, which made finding a time for debriefing difficult. Finally, the observers themselves did not want to be a source of additional interruptions. It should be noted that the IMPACTT observation methodology differs from that of McCurdie *et al.* in that it was developed to meet the objectives of the QualiREL Santé support system. In the DPM, interruptions are contextualized by asking professionals to identify the extent to which measures are required to avoid interruptions, and to help them to respond to requests requiring their attention. In the framework of the IMPACTT study, the results of our observations are an integral part of the support system. Our findings were presented during a joint meeting with the team, and should make it possible to look again at the tasks that the team wishes to protect from interruptions. The team can then choose which solutions should be implemented.

**Strengths and limitations.** Our study shows that the data collection guides reported in McCurdie *et al.* [27] can be adapted to different disciplines and other contexts. When transcribing the data, the observers rarely needed to discuss an interruption (a maximum of ten per team, results not presented). This was made possible by transcribing observations as soon as possible, to avoid memory bias. It was also made possible by the clear operational definition of an interruption [22], and the use of predefined response modes. The literature emphasizes that the clarity of the definition minimizes measurement errors or observer bias [23], and that predefined response modes help to optimize inter-observer reliability [23, 27]. It should also be noted that all observations were carried out over a short period of time; this was to ensure that there was no variation in how the same observer recorded interruptions (intra-observer reliability) [23].

However, it should be kept in mind that, as with any observational study, it is possible that the presence of observers affected the natural behavior of participants and, ultimately, our results [26]. To minimize this effect, the observers were acclimatized to the environment before beginning actual observations. In particular, they assured participating professionals that their performance would not be evaluated, and that all data would be recorded anonymously. Observers also paid close attention to where they were, with the aim of being far enough away from the professional so as not to interfere with their activities [23]. Another limitation concerns the absence of observers in the patient's room. Although some task interruptions could be quantified by observers, despite the fact that they were not in the room (e.g., another professional knocking on the door; a lack or failure of equipment that required the professional to leave the room), others could not be observed (e.g., when the professional in the room interrupted him or herself).

## Implications for research and teams

The implementation of the data collection guides, which we call in the following *Team'IT*, using the IMPACTT method, was evaluated by different pairs of observers. Its validation is currently in preparation for publication.

Beginning with the implications for research, we were able to collect a considerable corpus of data to characterize interruptions. Our results are promising in terms of their scientific contributions. Observational audits of interruptions are typically used with the objective of counting them. The Team'IT system adds value, as it takes account of how the team functions when characterizing interruptions. Following the implementation of the DPM by McCurdie *et al.*

[27], a third of all interruptions that occurred in the audited ICU were related to an interaction between coordination and care delivery functions. These interactions have a direct application in terms of patient safety. The care delivery function is the safety barrier that is closest to the patient, and it must be given particular attention. The other functions (coordination and organization) must be anticipated upstream of care delivery, in order that patient safety does not fall solely on the shoulders of the so-called last-in-line actors.

As described by Salas *et al*. [37], assumptions about human relationships and interactions in a team in a specific environment, such as an organization, will directly influence behaviors related to care safety. These aspects contribute to the question of the avoidability of interruptions. To date, nothing is known about the percentage of avoidable interruptions. We plan to publish an initial definition, and define the characteristics of the avoidability and inevitability of interruptions as part of this research project. The 23 teams that participated in the project will be able to use these results to identify interventions with a view to limiting their interruptions.

Turning to the implications for care teams, QualiREL Santé would like to publish the data collection methodology. The research team has begun to think about which items could, and would, be interesting in a non-research context. Practical advice for implementing the methodology has been formalized and is in the process of being published. This advice relates to preparations for observations (training of observers), the collection of data related to interruptions (observer behavior, observation of certain activities and professionals according to the time of day, recommendations for specific observations of pluri-professional activities), and data transcription. In the aftermath of the COVID crisis and the heavy demands it has placed on teams, the Team'IT method is a way to reexamine interruptions and team dynamics. In France, the latest HAS recommendations, published in the context of the certification of establishments, emphasize the need for teamwork, which is seen as the driving force behind improvements in professional practice [38].

The present study is part of the literature that investigates the challenges associated with the organization of hospital care, and cooperation between professionals, with the ultimate objective of providing safe, high-quality care. Our approach seeks to prevent disruption to the professional activities that make up part of the patient's care pathway (a hospital department). It would be interesting to link our work to research at the scale of the patient's overall care pathway (in particular, the outpatient pathway) by adopting a process structuring approach, and to integrate other themes such as pricing and the cost of a lack of quality [1, 39]. The findings from such studies could be formulated in terms of the flow and effectiveness of pathways, and help to improve the design of healthcare systems [2].

## Conclusion

In France, healthcare teams have struggled to address the problem of interruptions. Therefore, the new observational approach, the DPM, examines them from the point of view of the work functions that make up the system, and appears to be an opportunity to reflect, as a team, on the management of interruptions. We adapted the DPM to inpatient hospital care in France. The data that is collected using the observational tool, and the corresponding response modes were tailored to take into account the French definition of interruptions. We believe that our categorization of response modes is exhaustive. The terminology describing work functions was clarified. The period of observation was well-received by participating teams. These observations represent the first step in a support system for teams. Developed by QualiREL Santé, this system will enable them to examine their work practices and the avoidability of interruptions. This study is part of an approach that seeks to improve professional practice and risk

management. Our work is part of an approach that seeks to improve professional practices and risk management. Our results contribute to the longstanding, complex debate about the flow and effectiveness of patient pathways, leading, in the long term, to improvements in healthcare systems.

## Supporting information

**S1 File. Extended, pre-experimental version.**
(DOCX)

## Acknowledgments

The authors would like to thank all 23 teams who took part in the IMPACTT project, as well as Valérie De Salins and Céline Poulain for their participation in data collection. Thanks also go to Tara McCurdie for sharing the DPM guides, and to Elaine Seery for translation services.

## Author Contributions

**Conceptualization:** Delphine Teigné, Guillaume Mabileau, Noémie Terrien.

**Data curation:** Delphine Teigné, Guillaume Mabileau.

**Formal analysis:** Delphine Teigné, Guillaume Mabileau.

**Investigation:** Delphine Teigné, Lucie Cazet, Noémie Terrien.

**Methodology:** Delphine Teigné, Noémie Terrien.

**Project administration:** Delphine Teigné.

**Resources:** Delphine Teigné, Noémie Terrien.

**Supervision:** Noémie Terrien.

**Validation:** Delphine Teigné, Lucie Cazet, Guillaume Mabileau, Noémie Terrien.

**Visualization:** Delphine Teigné, Guillaume Mabileau.

**Writing – original draft:** Delphine Teigné, Noémie Terrien.

**Writing – review & editing:** Delphine Teigné, Lucie Cazet, Guillaume Mabileau, Noémie Terrien.

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
