## [Decision Letter · Decision Letter 0]

19 Oct 2022

PONE-D-22-18220Task interruptions from the perspective of work functions: the development of an observational tool applied to inpatient hospital care in France

The Team’IT toolPLOS ONE

Dear Dr. Terrien,

Thank you for submitting your manuscript to PLOS ONE. After careful consideration, we feel that it has merit but does not fully meet PLOS ONE’s publication criteria as it currently stands. Therefore, we invite you to submit a revised version of the manuscript that addresses the points raised during the review process.

 As you will see from the comments of the reviewers, they suggest the integration of more literature/references into the introduction and conclusion to better build and clarify your case, plus to improve the methods section so that readers quickly find the relevant information. 

We look forward to receiving your revised manuscript.

Kind regards,

Elisabeth Nöhammer

Academic Editor

PLOS ONE

Journal Requirements:

d)        If you did not receive any funding for this study, please state: “The authors received no specific funding for this work.

Reviewers' comments:

Reviewer's Responses to Questions

**Comments to the Author**

1. Is the manuscript technically sound, and do the data support the conclusions?

Reviewer #1: Yes

Reviewer #2: Yes

2. Has the statistical analysis been performed appropriately and rigorously? 

Reviewer #1: Yes

Reviewer #2: Yes

3. Have the authors made all data underlying the findings in their manuscript fully available?

Reviewer #1: Yes

Reviewer #2: Yes

4. Is the manuscript presented in an intelligible fashion and written in standard English?

Reviewer #1: Yes

Reviewer #2: Yes

5. Review Comments to the Author

Reviewer #1: Thank you for the opportunity to review the manuscript “Task interruptions from the perspective of work functions: the development of an observational tool applied to inpatient hospital care in France The Team’IT tool” (PONE-D-22-18220).

The authors developed an observational tool to characterize interruptions from the point of view of work functions that is tailored to French hospital units providing inpatient care. The aim was to adapt items recorded using the DPM and their response categories, and to study the acceptability of observing interruptions for participating teams.

The subject of the paper is interesting. The topic is also important in the international context. However, some parts of the manuscript must be carefully revised.

One of my concerns related to the introduction literature. Greater details about previous studies results are needed than currently provided that builds the case for having conducted the current study. This could also strengthen the discussion, as it is quite common to refer to findings from those studies relative to the current findings in the discussion and conclusions sections. In this context, clinical pathway (cp) management should also be included. Deviations from the treatment path are often indications of delays and interruptions. This is important because many clinics work with cp and co-cp and can thus generate structured analysis to identify problems.

The methods part is too detailed and should be shortened to include relevant information. In order to make this part clearer for the reader, a table or graphic should summarize the essential steps.

Conclusion: The implications for global health services should be worked out better.

Reviewer #2: The aim of the present study was to develop an instrument suitable for collecting observational data on the characteristics of interruptions, especially with regard to work functions. In doing so, the study builds on the DPM developed by Mc Curdie and was extended to include additional items.

Especially at the beginning of the study, it would be good to have a more detailed discussion of the term "interruption." Does interruption refer to core processes of medical performance? Is it process interruptions? Leaving this out could be a bit confusing for the reader, especially at the beginning. Especially in reference to line 228. The observers did not enter the patients' rooms, but that is where many interruptions occur (phone calls, patient questions, etc.)

At the end of the study, these aspects are again presented more clearly (just through the results table) and are much easier to follow. A reference to clinical processes would nevertheless be worth considering in this context.

The description of the inclusion criteria ( from line 141) leave room for interpretation. It is advisable to define here exactly what is meant by practicing a specialty. Does it apply to all healthprofessionals? Only medical professionals?

It would also be worth considering a reference for further research needs or further investigations.

6. PLOS authors have the option to publish the peer review history of their article (what does this mean?). If published, this will include your full peer review and any attached files.

Reviewer #1: **Yes: **PD Dr. Dr. Tobias Romeyke

Reviewer #2: **Yes: **Natasa Neuhold

---

## [Author Response · Author response to Decision Letter 0]

22 Dec 2022

General comments

Below we set out our point-by-point responses to the reviewers’ feedback:

Reviewer #1

• One of my concerns related to the introduction literature. Greater details about previous studies results are needed than currently provided that builds the case for having conducted the current study. This could also strengthen the discussion, as it is quite common to refer to findings from those studies relative to the current findings in the discussion and conclusions sections. In this context, clinical pathway (cp) management should also be included. Deviations from the treatment path are often indications of delays and interruptions. This is important because many clinics work with cp and co-cp and can thus generate structured analysis to identify problems.

Thank you for your comment. It is indeed important to situate our study within the broader context, notably with respect to the various methods and tools that are used to evaluate professional practices and risk management. As suggested, we have now added (in the Introduction) further details regarding how the method presented in our article (based on process analysis and failure mode analysis) can be related to the clinical pathway method (lines 59 to 67 – second paragraph and beginning of the third).

It is clear that many different methods and tools have been developed to evaluate professional practices, and can be used to analyze and identify patient risks. The clinical pathway method, in particular, supports the study of the key elements of coordination and cooperation in the patient’s care pathway. On the other hand, process analysis and failure mode analysis both seek to prevent disruption to professional activities at certain points along the patient’s care pathway. Thus, our study adopts the perspective of risk management associated with care, and ensuring the safety of a part of the care pathway (corresponding to the studied hospital department).

Furthermore, as suggested, we have strengthened our Discussion (lines 435 to 443, last paragraph), Conclusion (lines 456 to 459, last 2 sentences) and the Abstract (lines 45 to 47, last sentence) by situating our work with respect to:

- the challenges associated with the organization of hospital care, and cooperation between professionals, with the ultimate objective of providing safe, high-quality care,

- the patient’s overall care pathway.

Finally, we note the value of linking our work to the study of other themes, such as pricing and the cost of a lack of quality when studying the flow and effectiveness of care pathways. Such knowledge will help to improve the design of healthcare systems.

• The methods part is too detailed and should be shortened to include relevant information. In order to make this part clearer for the reader, a table or graphic should summarize the essential steps.

Thank you for your comment. We agree that the detailed presentation of the pre-experimental version was confusing for the reader. We have now moved this to the Supplementary Material (Appendix 1), where the interested reader can refer to it. The Method section has consequently been shortened by 324 words. Following this modification, we feel that it is not particularly helpful to add a table or figure to summarize the key steps. However, we remain open to further suggestions from the reviewer.

• Conclusion: The implications for global health services should be worked out better.

Please see our response to your first point. The Conclusion has now been strengthened to take your comment into account.

Reviewer #2

• Especially at the beginning of the study, it would be good to have a more detailed discussion of the term "interruption." Does interruption refer to core processes of medical performance? Is it process interruptions? Leaving this out could be a bit confusing for the reader, especially at the beginning. Especially in reference to line 228. The observers did not enter the patients' rooms, but that is where many interruptions occur (phone calls, patient questions, etc.) At the end of the study, these aspects are again presented more clearly (just through the results table) and are much easier to follow. A reference to clinical processes would nevertheless be worth considering in this context.

Thank you for your comments and suggestions.

Reviewer 1 made a similar comment, and we have therefore added further details to the Introduction about how our method (based on process analysis and failure mode analysis) relates to the clinical pathway method (lines 59 to 67 - second paragraph and beginning of the third). Our method is one of many approaches that can be used to prevent disruption to the professional activities that make up the patient’s care pathway (in our case, the studied hospital department). The clinical pathway method studies key elements in the coordination and cooperation of the patient’s pathway.

The definition used in France, provided by the HAS, specifies the key terms used in our approach: “[the] unexpected, temporary or definitive cessation of a human activity” (lines 77 to 82). The definition given by Speier et al. (1999) also specifies that it is an event that interferes with the performance of a specific task (lines 75 to 77). To improve clarity, at the beginning of our Discussion, we reiterate that task interruptions are associated with disruption to the activity of professionals (line 330). 

Concerning interruptions that occurred in the patient’s room, we wished to avoid placing observers in the room in order to respect the dignity of the patient, and to ensure that the delivery of care was not impacted. This point was specified in our observation protocol, and was integral to ethical approval being granted. However, some task interruptions could be quantified by observers, despite the fact that the person was not in the room (e.g., another professional knocking on the door; equipment failure that required the professional to leave the room; a ringing telephone). Nevertheless, it is true that other task interruptions could not be observed (e.g., when the professional in the room interrupted him or herself). Although observers made a point of asking the observed professional, at the end of the observation, about any task interruptions that had occurred in the room, the latter’s point of view could be subjective, because he or she was unaware of the importance of the question. This bias has been added to the Discussion section (lines 394 to 399). 

We hope that this new material responds to your concerns. Nevertheless, we remain open to any other suggestions.

• The description of the inclusion criteria (from line 141) leave room for interpretation. It is advisable to define here exactly what is meant by practicing a specialty. Does it apply to all health professionals? Only medical professionals? It would also be worth considering a reference for further research needs or further investigations.

Thank you for your comment. We agree that further clarification is necessary.

First, we thought it appropriate to define in the Introduction (lines 120 to 123) the term “conventional hospitalization sector”. As suggested, we have added a reference. According to the French DREES (Direction de la recherche, des études, de l'évaluation et des statistiques), these sectors correspond to those in which a patient’s stay in a healthcare facility is longer than one day, and which requires accommodation for at least one night [1]. We have also specified in the Method (Definition of the team) that a care specialty is delivered by healthcare professionals (line 147), and that the IMPACTT research program targeted medicine or surgery (excluding intensive care) [1] (line 149). Consequently, we have reformulated and clarified our inclusion criteria. They are now as follows: “Inclusion criteria for the IMPACTT project were as follows: a) being located in the Pays de la Loire region (western France); b) being willing to implement a study on the management of interruptions; and c) being able to receive patients for a stay of more than one day, and requiring accommodation of at least one night, whose care is related to medicine or surgery (excluding intensive care)” (lines 168 to 172).

1. Directorate for Research, Studies, Evaluation and Statistics. Hospitalisation conventionnelle [inpatient hospital] [Internet]. [cité 21 nov 2022]. Disponible sur: https://drees.solidarites-sante.gouv.fr/hospitalisation-conventionnelle

---

## [Decision Letter · Decision Letter 1]

22 Feb 2023

Task interruptions from the perspective of work functions: the development of an observational tool applied to inpatient hospital care in France

The Team’IT tool

PONE-D-22-18220R1

Dear Dr. Terrien,

We’re pleased to inform you that your manuscript has been judged scientifically suitable for publication and will be formally accepted for publication once it meets all outstanding technical requirements.

Kind regards,

Elisabeth Nöhammer

Academic Editor

PLOS ONE

Additional Editor Comments (optional):

Reviewers' comments:

Reviewer's Responses to Questions

**Comments to the Author**

1. If the authors have adequately addressed your comments raised in a previous round of review and you feel that this manuscript is now acceptable for publication, you may indicate that here to bypass the “Comments to the Author” section, enter your conflict of interest statement in the “Confidential to Editor” section, and submit your "Accept" recommendation.

Reviewer #1: All comments have been addressed

Reviewer #2: All comments have been addressed

2. Is the manuscript technically sound, and do the data support the conclusions?

Reviewer #1: Yes

Reviewer #2: Yes

3. Has the statistical analysis been performed appropriately and rigorously? 

Reviewer #1: Yes

Reviewer #2: Yes

4. Have the authors made all data underlying the findings in their manuscript fully available?

Reviewer #1: Yes

Reviewer #2: Yes

5. Is the manuscript presented in an intelligible fashion and written in standard English?

Reviewer #1: Yes

Reviewer #2: Yes

6. Review Comments to the Author

Reviewer #1: The authors accepted and implemented the expert's suggestions and were thus able to improve the quality of the paper.

Reviewer #2: Adequate measures were derived from the aforementioned potential improvements and embedded in the article accordingly. Thus, there are no further objections in the course of the publication of this article.

7. PLOS authors have the option to publish the peer review history of their article (what does this mean?). If published, this will include your full peer review and any attached files.

Reviewer #1: No

Reviewer #2: **Yes: **Natasa Neuhold

---

## [Editor Report · Acceptance letter]

27 Feb 2023

PONE-D-22-18220R1 

Task interruptions from the perspective of work functions: the development of an observational tool applied to inpatient hospital care in France
The Team’IT tool 

Dear Dr. Terrien:

I'm pleased to inform you that your manuscript has been deemed suitable for publication in PLOS ONE. Congratulations! Your manuscript is now with our production department. 

Kind regards, 

on behalf of

Dr. Elisabeth Nöhammer 

Academic Editor

PLOS ONE